# Information Processing Speed Assessed with Letter Digit Substitution Test in Croatian Sample of Multiple Sclerosis Patients

**DOI:** 10.3390/diagnostics12010111

**Published:** 2022-01-04

**Authors:** Ana Jerković, Meri Matijaca, Ana Proroković, Anđela Šikić, Vana Košta, Ana Ćurković Katić, Krešimir Dolić, Klaudia Duka Glavor, Joško Šoda, Zoran Đogaš, Maja Rogić Vidaković

**Affiliations:** 1Laboratory for Human and Experimental Neurophysiology (LAHEN), Department of Neuroscience, School of Medicine, University of Split, Šoltanska 2, 21000 Split, Croatia; anasuto@gmail.com (A.J.); zdogas@mefst.hr (Z.Đ.); 2Department of Neurology, University Hospital of Split, Spinčićeva 1, 21000 Split, Croatia; meri.matijaca@gmail.com (M.M.); vanakosta@gmail.com (V.K.); ana.curkovic.katic@gmail.com (A.Ć.K.); 3Department of Psychology, University of Zadar, Ul. Mihovila Pavlinovića, 23000 Zadar, Croatia; aprorok@unizd.hr (A.P.); andela.sikic@gmail.com (A.Š.); 4Department of Radiology, University Hospital of Split, Spinčićeva 1, 21000 Split, Croatia; kdolic79@gmail.com; 5General Hospital Zadar, Neurology, Ul. Bože Peričića 5, 23000 Zadar, Croatia; klaudia.dukaglavor@gmail.com; 6Signal Processing, Analysis, Advanced Diagnostics Research and Education Laboratory (SPAADREL), Faculty of Maritime Studies, University of Split, Ul. Ruđera Boškovića 37, 21000 Split, Croatia; jsoda@pfst.hr; 7Split Sleep Medical Centre, University Hospital of Split, 21000 Split, Croatia

**Keywords:** multiple sclerosis, cognition, cognitive measure, information processing speed, Letter Digit Substitution Test

## Abstract

Cognitive impairment is a common complaint in people with multiple sclerosis (pwMS). The study objective was to determine the psychometric properties of the letter digit substitution test (LDST) that measures information processing speed and to investigate the impact of relevant predictors of LDST achievement in pwMS. The design was cross-sectional. The study included 87 pwMS and 154 control subjects. The validity of LDST was examined, and a hierarchical regression model was used to explore relevant predictors of LDST success. The LDST had excellent construct validity, as expressed by differences between pwMS and control subjects. Convergent validity of the LDST was supported by a significant moderate correlation with the expanded disability status scale (EDSS) (ρ = −0.36; *p* < 0.05) and a significantly strong correlation with the multiple sclerosis impact scale (MSIS-29) physical subscale (r = −0.64; *p* < 0.01). The LDTS score well differentiated the pwMS considering age, education, EDSS, disease duration, comorbidity, and medication therapy. Using the LDST as a criterion variable in pwMS results showed consistent evidence for the age, education, and EDSS impact on LDST performance. The best cut-off score of ≤35 discriminated the control and MS group. LDST proved to be a valid test for assessing information processing speed in pwMS.

## 1. Introduction

Multiple sclerosis (MS) is a chronic inflammatory neurological disease of the central nervous system with autoimmune etiology leading to a broad and complex clinical picture affecting 2–144 per 100,000 people in Japan, America, and Europe [1]. Clinical symptoms of MS include disturbances in motor functions (e.g., tremor, weakness, spasticity), sensory deficits (e.g., pain), visual impairments (e.g., diplopia and optic neuritis), vascular dysfunctions, obesity, cognitive impairments (e.g., attention deficits, working memory impairments, information processing), and mood disorders (e.g., depression, anxiety, stress). Approximately 43–70% of people with MS (pwMS) have a cognitive impairment, with a prevalence of 20–50% of pwMS having an impairment in information processing speed [2,3]. The impairment associated with information processing is the first cognitive deficit to emerge in pwMS [4]. In clinical practice, information processing speed is frequently assessed via the symbol/digit substitution test (the participant is instructed to associate symbols to letters (or vice versa) or compare symbols and provide an oral or written response (e.g., symbol digit modalities test (SDMT)) [5]. There are several adaptations of the SDMT test, and the most commonly used test in pwMS is the one where numbers replace the given symbols (SDMT) [5]. In the Croatian clinical setting of pwMS, two tests are randomly used for assessing information processing speed, SDMT [5] and the letter digit substitution test (LDST) [6,7], an adaptation of earlier substitution tests, the digit symbol substitution test (DSST) [8], and the SDMT [5,9].

To date, the psychometric properties of LDST have not yet been determined in pwMS. In this study, we aimed to evaluate the psychometric properties of the LDST [6,7] in terms of validity and reliability in a Croatian sample of pwMS and including a non-clinical sample (control healthy subjects). In addition, using LDST as criterion variables, the study aim was to investigate the impact of relevant demographic and disease-related predictors on LDST achievement in pwMS.

## 2. Materials and Methods

### 2.1. Study Population

A total of 87 pwMS and 154 control subjects were included in the study. The pwMS were recruited from November 2020 to April 2021 at two locations: The Neurology Department of the University Hospital of Split and the School of Medicine of the University of Split. All of the participants were addressed in personal communication, were contacted by e-mail or mobile phone, and were approached during their usual clinical care at the Department of Neurology. The pwMS recruited through the Association for Multiple Sclerosis Society of Croatia (AMSSC) were tested at the School of Medicine. Inclusion criteria were as follows: (1) Age 18 or older; (2) fluent in Croatian; (3) able to provide informed consent to all of the procedures; exclusion criteria were: (1) History of neurological disorder other than MS; (2) history of psychiatric disorder; and (3) history of the developmental disorder (e.g., learning disability).

Since the study was conducted during the COVID-19 disease, the physical contact between the examiner and the participant was limited by having the general and MSIS-29 [10,11] questionnaires sent via Google questionnaire forms, which were accessed by the participant via a link. The participants were instructed to complete the questionnaires sent via e-mail on the same day when LDST was administered. Eleven participants who were either older or not using technology completed the paper version of the questionnaires.

In the group of pwMS, 82% were female with a mean age of 42.7 ± 12.5 years, and 18% were men with a mean age of 42.1 ± 10.9 years. Most of the people with MS were right-handed (93.1%) and between 35 to 60 years old (62%). Most of the pwMS completed high school (66%), graduate study (21.1%), undergraduate study (9.4%), and primary school (3.5%). Most of the pwMS were diagnosed with MS disease between 6 to 11 years (43.5%), 32.2% were diagnosed between 0 to 5 years, and 24.3% reported over 11 years of MS diagnosis. The mean duration of the MS disease for all of the pwMS was 9.1 ± 7.3. A majority of the subjects had relapsing-remitting MS (RRMS) (79%), while others reported having secondary progressive MS (SPMS) (2.5%) and primary progressive MS (PPMS) (9.2%). Certain people with MS (6.8%) did not provide information on the type of MS, and in 2.5% of pwMS, the MS type was not established. The median expanded disability status scale (EDSS) score for people with MS was 1.0 ± 2.5. Of 87 pwMS, 36% had comorbidities, including endocrine, nutritional, and metabolic diseases (33%) and diseases of the respiratory system (16%). Three MS subjects were in wheelchairs. A total of 51% of pwMS were treated with immunomodulatory drugs. Most of the pwMS were treated with glatiramer-acetate (31%), teriflunomide (15.5%), and dimethyl fumarate (15.5%) drugs.

In the control group, 70% of participants were women with a mean age of 43.8 ± 13.3 years, and 30% were men with a mean age of 45.0 ± 16.2 years. Most of the control subjects were right-handed (91.5%) and between 35 to 60 years old (53%), and most of them completed high school (44.2%), graduate study (37.6%), undergraduate study (14.3%), and primary school (3.9%). Of 154 control subjects, 27% had comorbidities, of which the most common were diseases of the circulatory (48%) and endocrine, nutritional, and metabolic diseases (27%). Twenty-three percent (23%) of the control subjects took medications. Most of the control subjects took antihypertensive (36%) and pain medications (12%).

### 2.2. Questionnaires

#### 2.2.1. General Questionnaire

The general questionnaire contained: Demographic data (age, sex, handedness), education level, comorbidity, medication intake related to comorbidities, and MS-related information, including duration of the disease, MS type [12], EDSS score [13], and information on medication intake related to the MS treatment.

#### 2.2.2. Letter Digit Substitution Test (LDST)

The LDST is an adapted substitution test [5,6,7,8] developed to measure information processing speed. At the top of the test is a key with nine letters of the alphabet and the corresponding numbers from 1 to 9 presented on an A4 paper that measures 210 × 297 mm or 8.27 × 11.69 inches. The A4 paper is an International/European paper size established by the ISO, the International Standards Organization. Below the key is a table of randomly arranged letters, and the participants are required to replace the randomized letters with the appropriate digit indicated by the key. The first 10 places in the table are used to ensure an understanding of the instructions and the correct solving of the test. After completing these items, participants are instructed to replace the remaining items as quickly as possible (within 60 s) [6], from left to right, without skipping the empty places in the table. The dependent variable represents the correct number of filled squares within 60 s. The key and the stimuli are the same for the oral and written versions of the LDST. In the present study, the subjects completed a written version of the LDST test. The LDST test sheet is given in the Appendix A (Appendix A).

#### 2.2.3. Multiple Sclerosis Impact Scale (MSIS-29)

The multiple sclerosis impact scale (MSIS-29) [10] is a self-report scale measuring the psychological and physical impact of the MS disease on the patient. The scale is structured into two subscales, a 20-item scale for measuring the physical impact and a 9-item scale for measuring the psychological impact of the disease in the past 2 weeks. For each statement, the participant circles the number that best describes his/her condition on a five-point Likert scale (1 = not at all, 2 = a little, 3 = moderately, 4 = quite a bit, and 5 = extremely). The MSIS-29 score is generated by summing the scores on two subscales, with higher scores indicating a more severe disease burden. The MSIS-29 validated to the Croatian MS population was used in the present study [11].

### 2.3. Validation Procedure

The internal consistency of the MSIS-29 scale was estimated by Cronbach’s alpha coefficients and inter-item correlations. The convergent validity of the LDST test was demonstrated by the correlation between LDST and MSIS-29 subscales and with EDSS. The concurrent validity was evaluated with comparisons of LDST scores between pwMS and control subjects. The receiver operating characteristic curve analysis (ROC) using the Youden Index and the area under the curve (AUC) was performed to determine the optimal cut-off point discriminating control group and MS group. The impact of relevant predictors on LDST achievement was investigated by the hierarchical regression model.

### 2.4. Statistical Analysis

Shapiro-Wilk normality test was used to validate the assumption of normality and showed no departs from a normal distribution for LDST. Furthermore, parameters of skewness and kurtosis indicated acceptable values for the parametric statistic. Nonparametric statistics were used when normality assumptions were violated. Mean value comparisons between relevant demographic data, test, and scale variables were carried using t-tests, ANOVA, Chi-square, Mann Whitney U test, and Kruskal-Wallis test. The post hoc Fisher LSD test was calculated when using multiple comparisons. Levene’s test was used to assess the equality of variances between groups. Correlation analyses were conducted using Pearson’s R coefficient and Spearman rank-order correlation (ρ). The multiple regression analysis was performed to estimate the influence of age, education, duration of the disease, comorbidity status, and EDSS on LDST performance. The results were expressed as multiple R and beta coefficients. Descriptive statistics of relevant participants’ characteristics and applied self-report scales and LDST were summarized by N, percentage, mean and standard deviations, median and interquartile range (IQR). A threshold of *p* < 0.05 was used for the determination of the level of effect significance. Data analysis was performed using the software Statistica 12.

## 3. Results

### 3.1. Overview Results

The demographic characteristics, MS-related variables, and the mean results on the LDST and MSIS-29 of pwMS and control subjects are presented in Table 1. No significant sex (χ^2^ = 3.5, *p* = 0.06, *p* > 0.05), age (t = 0.9, df = 239, *p* > 0.05), and education (Z = 0.61, *p* = 0.51, *p* > 0.05) differences were found between pwMS and control subjects.

### 3.2. Psychometric Properties of the MSIS-29

Expressed by Cronbach’s α coefficients, both MSIS-29 subscales (αMSIS-PHYS = 0.95 to αMSIS-PSY = 0.94) had an excellent internal consistency. Values of Cronbach’s α for MSIS-29 scales are considered indicative of good reliability. Inter-item correlations for MSIS-29 scales were >0.4, indicating that all of the items on each subscale correlate very well with the scale overall.

### 3.3. Construct Validity of the LDST

Control subjects achieved a significantly better performance on the LDST (t = 4.4, *p* < 0.01, df = 239) in comparison to pwMS (Figure 1).

Furthermore, the achievement on the LDST differentiates pwMS concerning education and age. Respectively, significant differences were found between pwMS age groups of 19–34 y, 35–60 y, 60–73 y (H = 26.2, *p* < 0.001) in LDST performance, indicating that the performance on LDST decreases in the function of age (Figure 2). Moreover, a significant negative correlation between age and LDST score indicates that the older pwMS had poorer performance on LDST (Figure 2, Table 2). Significant differences in LDST achievement regarding the education level were also observed (H = 15.3, *p* < 0.01) (Figure 2). The higher the education level, the better the performance on the LDST in pwMS. A significant correlation between education and LDST score confirms this finding in pwMS (ρ = 0.41*, *p* < 0.05).

A significantly higher LDST performance was found in pwMS with partially preserved mobility (EDSS 0–4.5) concerning pwMS with partially or fully impaired mobility (5–9.5) (Z = 1.96, *p* < 0.05) (Table 2, Figure 2). Furthermore, a significantly lower LDST performance was found in pwMS with comorbidity regarding pwMS with no comorbidity (t = 1.99, *p* < 0.05) (Table 2, Figure 2). Moreover, duration of MS disease was found to be a significant variable for LDST performance (F = 10.8, *p* < 0.001) (Table 2, Figure 2). People that have MS for a longer period show a decrease in LDST achievement (*p*_0 to 5 y vs._
_6 to 11 y_ < 0.01; *p*_0 to 5 y vs._
_over 11 y_ < 0.01; *p*_6 to 11 y vs_. _over 11 y_ < 0.01) (Table 2, Figure 2). Furthermore, a significantly better LDST performance was found in pwMS that were using immunomodulatory drugs in regards to pwMS that did not use immunomodulatory drugs (t = 3.3, *p* < 0.01) (Table 2, Figure 2). No differences were found in LDST scores in regards to MS type (H = 6.5, *p* > 0.05) and sex (t = 0.22, *p* > 0.05) (Table 2).

Table 3 and Figure 3 present the ROC analysis using the Youden Index and the AUC to determine the optimal cut-off point discriminating control group and MS group. The highest value of the Youden Index (J = 0.281) was obtained for a cut-off point of ≤35. The statistically significant AUC was 0.698 (*p* < 0.0001) with 95% confidence interval 0.677 to 0.718 (Figure 3). Both parameters (J and AUC) indicate that the LDST has a satisfactory diagnostic validity for group differentiation with an optimal criterion of ≤35 (Table 3).

### 3.4. Convergent and Divergent Validity of the LDST

LDST achievement is correlated with MSIS-29 subscales, with the correlations of LDST higher with the physical MSIS-29 subscale (r = 0.64, *p* < 0.01) compared to the psychological MSIS-29 subscale (r = −0.40, *p* < 0.01) (Table 4). Furthermore, the convergent validity of the LDST is supported by a significant moderate correlation with EDSS (ρ = −0.36, *p* < 0.05) (Table 4).

### 3.5. Multiple Regression Analysis

Table 5 represents the results of multiple hierarchical regression analyses for the age, education, comorbidity status, duration of the disease, and EDSS score on LDST performance of pwMS. In the first set of predictor variables, age and education had significant β and explained 41% of LDST variance. In step 2, which included the EDSS score, comorbidity status, and disease duration, only the EDSS score had significant β. However, the contribution of the second set of variables is not significant, and the percentages of explained variance did not increase. Age, education, and EDSS significantly contributed to the explanation of the LDST performance in pwMS.

## 4. Discussion

The present study examined the psychometric properties of the LDST, revealing its excellent construct validity on a Croatian sample of pwMS. The LDST score for pwMS was lower than for the control subjects, indicating more pronounced information processing impairments associated with the MS disease. The convergent validity of the LDST is supported by a significantly strong correlation with the physical MSIS-29 subscale and a significant moderate correlation with the EDSS. The LDTS score well differentiated pwMS considering age, education, EDSS, duration of the MS disease, comorbidity status, and usage of immunomodulatory therapy. Age, education, and EDSS were the most important predictors of LDST performance of pwMS. LDTS performance is strongly influenced by age, education, and sex in the general healthy subjects [6]. However, in the present study, sex was not found to predict LDST performance in pwMS. Results also indicate that the best cut-off LDST score of ≤35 has a satisfactory diagnostic validity for discriminating the control and MS group. Therefore, LDST proved to be a valuable, valid, and reliable tool for assessing information processing speed in pwMS, differentiating pwMS from the control subjects. Furthermore, LDST has been shown to be a culturally robust test [14], and previous studies included cognitively intact, healthy people to generate normative data of the LDST. The usefulness of the LDST has been previously demonstrated in different clinical samples [15,16,17].

The present study has limitations that need to be considered. A possible limitation would be the time of conducting the survey. Namely, the study was conducted during the COVID-19 pandemic when social distancing was reduced and possibly influenced the psychological functioning of pwMS. However, we assume that COVID-19 did not significantly affect the LDST performance in pwMS. A comparison with previous studies shows similarities in the prevalence of psychological disturbances in the pwMS population independently of external factors unrelated to MS [18]. The second criticism of the present study might be the reason the SDMT test measuring information processing speed was not validated on Croatian MS samples rather than LDST. Moreover, SDMT is included in the battery tests, such as the Brief International Cognitive Assessment for MS (BICAMS) proposed by Langdon et al. [9]. First, we considered the validation of LDST since it was previously performed in healthy subjects [19]. Additionally, due to the fact that both tests, SDMT and LDST, are randomly used by psychologists in Croatian hospitals to assess information processing speed in pwMS. Second, to date, no study compared the LDST and SDMT as two similar versions of tests that measure information processing speed in pwMS. Van der Elst et al. [6] problematized the SDMT versus LDST, stating that in SDMT, the participants must first learn the abstract symbols to pair them properly. Therefore, the question is whether the result is information processing speed, learning, and memory ability or some other cognitive skill. As the name suggests, in LDST, the letters are paired with numbers. As a result, the LDST differs from the SDMT since it uses well-known and well-learned letters rather than abstract forms. In this way, to successfully solve the LDST test, the participant must remember only the association of an already well-known letter and number (e.g., the letter B is paired with the number 2). Moreover, LDST and SDMT differ in the required time to solve the test. After completing the first 10 items with guidance and with the purpose of practice, the subject is timed to determine how many responses can be made in 90 s in SDMT [20], while in LDST, the subject is given 60 s [6]. The second limitation refers to the smaller sample size for different phenotypes of MS in our study. This might be one of the reasons for the discrepancy in the results of our study and other studies that reported a higher prevalence of cognitive impairment in SPMS and PPMS [21].

The specific significance of this study is attributed to the contribution of the LDST validation for the population suffering from MS in the Croatian-speaking area and the ability of creating preconditions for the BICAMS validation [9,22,23]. The validation of BICAMS in pwMS has been performed in different countries [24,25,26,27,28,29,30,31,32,33,34,35,36,37], and inquires that authors tend to apply specific similar tests in their country if the existing proposed tests through the BICAMS battery (comprising of SDMT, California Verbal Learning Test-II, and the Brief Visuospatial Memory Test-Revised) were not available or could not be applied in their country for certain reasons [25,34,36]. Future studies might compare the LDST and SDMT in pwMS.

## 5. Conclusions

In conclusion, the LDST proved to be valid and reliable for assessing information processing speed in Croatian samples of pwMS. The validated LDST will serve the clinical practice and research as a diagnostic instrument for evaluating and monitoring information processing. Furthermore, the present study’s findings point to the clinical usefulness of the LDST application with EDSS and MSIS-29 for assessing the cognitive status, disability level, and impact of MS disease on physical and psychological functioning.

## Figures and Tables

**Figure 1 diagnostics-12-00111-f001:**
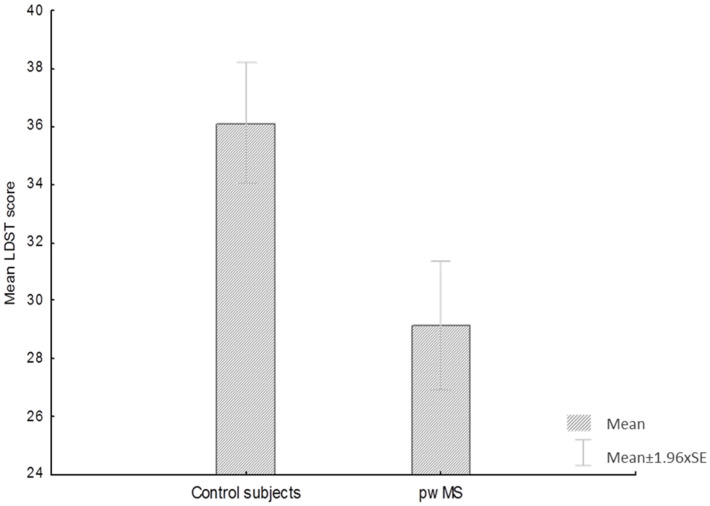
Box plots showing means and standard deviations of LDST in pwMS and control subjects.

**Figure 2 diagnostics-12-00111-f002:**
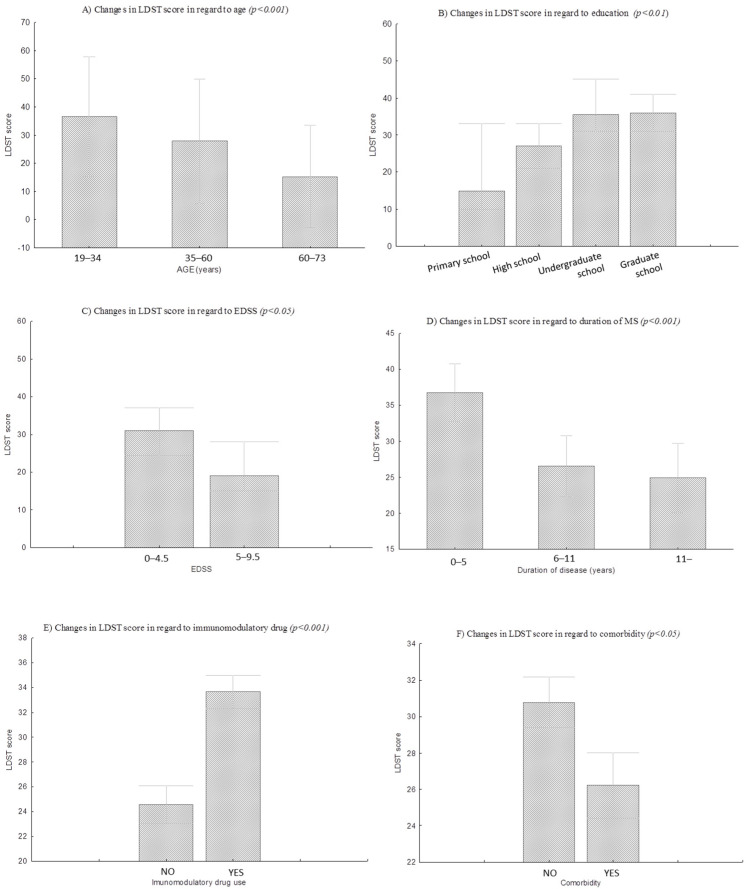
Changes in LDST performance regarding age, education, EDSS, duration of disease, comorbidity, and immunomodulatory drug use in pwMS.

**Figure 3 diagnostics-12-00111-f003:**
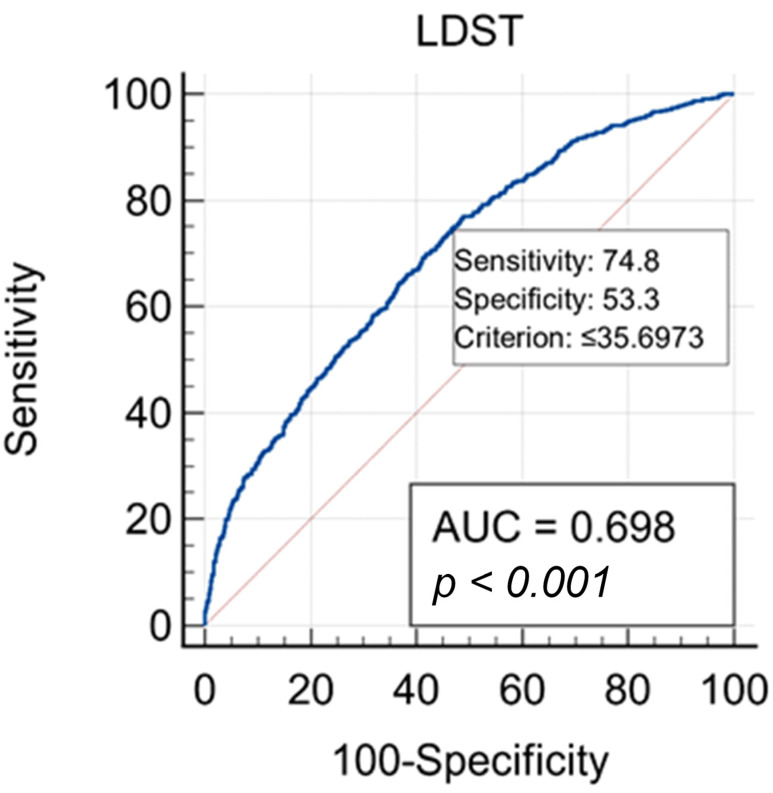
ROC curve of LDST.

**Table 1 diagnostics-12-00111-t001:** Characteristics of study participants.

	Control Subjects (N = 196)	People with MS (N = 87)	Test	*p*-Value
Age in years (mean ± SD) Female (mean ± SD) Male (mean ± SD)	44.2 ± 14.2 43.8 ± 13.3 45.0 ± 16.2	42.8 ± 12.2 42.7 ± 12.5 42.1 ± 10.9	t = 0.9	*p* = 0.37
Age (range)	18–81	19–73		
Sex			χ^2^ = 3.5	*p* = 0.06
Female	70%	81%		
Male	30%	18%		
EDSS (median ± IQR, range)		1.0 ± 2.5, 0–80–9		
EDSS ^†^		1.0 ± 2.0		
EDSS ^††^		6 ± 1.0		
LDST and self-report scale (mean ± SD)				
LDST	36.6 ± 10.2	29.2 ± 10.4	t = 5.2	*p* < 0.01
MSIS-29 PHYS		45.1 ± 18.0		
MSIS-29 PSY		22.1 ± 9.7		

Abbreviations: SD—standard deviation; IQR—interquartile range; EDSS—expanded disability status scale; EDSS ^†^—partially preserved mobility 0–4.5; EDSS ^††^—partially or fully impaired mobility 5–9.5; LDST—letter digit substitution test; MSIS-29-PHYS—physical subscale; MSIS-29-PSY—psychological subscale; *χ^2^*—Chi-squared test.

**Table 2 diagnostics-12-00111-t002:** Differences in LDST performance regarding relevant demographic and disease-related variables in pwMS.

		%	LDST (Mean ± SD)	Test	*p*-Value
Sex	Women	82.0	29.3 (10.4)	*p* = 0.22	>0.05
	Men	18.0	28.6 (10.6)		
Age (years)	19–34 years	28.7	36.5 (8.5)	H = 26.2	*p* < 0.001
	35–60 years	62.0	27.8 (8.9)		
	60–73 years	9.3	15.3 (7.3)		
Education	Primary school	4.3	19.3 (12.1)	H = 15.3	*p* < 0.01
	High school	65.5	26.8 (9.5)		
	Undergraduate studies	9.5	37.4 (7.6)		
	Graduate study	20.7	34.2 (10.2)		
EDSS	EDSS ^†^	85	29.8 (10.3)	Z = 1.96	<0.05
	EDSS ^††^	15	22 (9.1)		
Comorbidity	YES	64.4	26.2 (10.0)	t = 1.99	<0.05
	NO	35.6	30.8 (10.3)		
Duration of MS disease	0 to 5 years	32.2	35.6 (8.5)	F = 10.8	<0.001
	6 to 11 years	43.5	26.8 (9.3)		
	over 11 years	24.3	24.9 (10.8)		
MS type	RRMS	79.0	30.2 (10.2)	H = 6.5	>0.05
	SPMS	2.5	28.5 (4.9)		
	PPMS	9.2	20.3 (9.7)		
	Not known	6.8	26.6 (11.8)		
Immunomodulatory drug	YES	51.0	24.6 (9.9)	t = −4.5	<0.001
	NO	49.0	33.6 (8.9)		

Abbreviations: LDST—letter digit substitution test; SD—standard deviation; RRMS—relapsing-remitting multiple sclerosis; SPMS—secondary progressive multiple sclerosis; PPMS—primary progressive multiple sclerosis; EDSS—expanded disability status scale; EDSS ^†^—partially preserved mobility 0–4.5; EDSS ^††^—partially or fully impaired mobility 5–9.5.

**Table 3 diagnostics-12-00111-t003:** Psychometric properties of LDST at different cut-off scores (ROC analysis).

LDST Scores	Sensitivity	Specificity	+LR	−LR
≤15	9.40	98.60	6.71	0.92
≤20	20.90	95.40	4.54	0.83
≤25	35.10	86.70	2.64	0.75
≤30	53.90	71.90	1.92	0.64
≤35	71.50	55.60	1.61	0.51
≤40	86.10	36.20	1.35	0.38
≤45	94.20	22.30	1.21	0.26
≤50	97.90	9.60	1.08	0.22

+LR: Likelihood ratio for a positive result; −LR: Likelihood ratio for a negative result.

**Table 4 diagnostics-12-00111-t004:** The correlation coefficient for LDST, EDSS, and MSIS-29 in pwMS (N = 87).

	LDST	MSIS-29 PHYS	MSIS-29 PSY	EDSS
LDST	-	−0.64 **	−0.40 **	−0.36 *
MSIS-29 PHYS		-	0.76 **	0.48 **
MSIS-29 PSY			-	0.20
EDSS				-

Abbreviations:*—<.05; **—*p* < 0.01; LDST—letter digit substitution test; MSIS-29 PHYS—physical subscale; MSIS-29 PSY—MSIS-29 psychological subscale; EDSS—expanded disability status scale.

**Table 5 diagnostics-12-00111-t005:** Multiple hierarchical regression analysis for the impact of age, sex, education, and EDSS score on LDST performance of pwMS.

		LDST	
		Step1	Step2
Predictors		*β*	*β*
Step 1	Age	−0.50 **	−0.49 **
	Education	0.24 **	0.18
	R^2^	0.41 **	
Step 2	EDSS		−0.19 **
	Comorbidity		−0.03
	Duration of disease		−0.08
	R^2^		0.47 **
	ΔR^2^		0.046

Abbreviations: EDSS–expanded disability status scale; β–standardized regression coefficient; R^2^–coefficient of determination; ΔR^2^–change in the coefficient of determination; ** *p* < 0.01, CI = 98%.

## Data Availability

The data presented in this study are available on request from the corresponding author. The data are not publicly available due to privacy restrictions.

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
