# Peer review of "Information Processing Speed Assessed with Letter Digit Substitution Test in Croatian Sample of Multiple Sclerosis Patients"

_diagnostics, 2022, doi:10.3390/diagnostics12010111_

Round 1
Reviewer 1 Report
Very interesting article, well-written. I only have a few remarks particularly regarding results and discussion :
-You partially stated why you chose to validate in Croatian MS population LDST over SMDT. Why is LDST more often used in Croatia? In the Authors opinion, what are the advantages of LDST? could also be the differences in time administration (60sec for LDST - reported in row 126 - and 90sec for SDMT - DOI: 10.1177/1352458517690821)
-figure 2 reports changes in LDST performance regarding age, education, EDSS, duration of disease. You need to check results in order to understand if the reported histograms are p< or > 0.5.
-Table 2: p>0.5 is reported for EDSS subgroups whereas in row 213
-Table 4 reports a negative correlation (-0.64) whereas the abstract and results (row 33 and 234) report a positive correlation. Could the author clarify this discrepancy? If I understand correctly > MSIS reports a higher impact of MS on QoL. How do you explain that patients with higher cognitive performances experience a lower QoL ? this should be discussed in discussion as well.
-How do you explain that there are no differences with progressive phenotypes? In literature, it is reported that progressive phenotype experiences a higher prevalence of CI (also considering that normally SPMS are older and with higer EDSS than RRMS) (doi: 10.3389/fneur.2019.00261)
Once these points have been improved, the work should be published for an extension of knowledge in the field.
Author Response
Dear Reviewer,
the responses to your valuable comments are attached in the word document.
Sincerely,
Maja Rogić Vidaković, corresponding author

Reviewer 2 Report
The author study are valuable but number of subjects for data collections are low. Also data collection collected remotely from patients are not reliable.
Author Response

(The authors gave the same response as above.)

Round 2
Reviewer 1 Report
Dear Authors thank you for you revisions. I only have two small further remarks regarding comment 1 and comment 2 that i do not find properly addresed in the current version:
-do you confirm that the patient has sixty seconds to complete LDST as you marked in row 133? on the contrary, the patients has 90 sec for SDMT ( DOI: 10.1177/1352458517690821). This could be an important difference (i.e an advantage for LDST over SDMT) and should be pointed out in discussion -comment 2 was indicating that FIGURE 2 is not clear enough. At the moment , it is not clear from the histograms or the caption what is statistically significant and what not. The reader needs to cheak Table 2 in roder to understand figure 2. Please add in FIGURE 2 histograms or in Figure 2 caption P-values.
Once these 2 points have been improved, the work should be
published for an extension of knowledge in the field.
Author Response
Dear Reviewer, the author's answers are attached in the word document.
Thank you.
Corresponding author

Reviewer 2 Report
The author managed the revision very well.
Author Response
Authors response to Reviewer#2: We thank the honorable reviewer for the time spent reviewing our paper. Also, we are grateful for positive comments and suggestions, which helped us to improve the resubmitted paper significantly.
